# A Novel Adaptive Noise Elimination Algorithm in Long RR Interval Sequences for Heart Rate Variability Analysis

**DOI:** 10.3390/s22239213

**Published:** 2022-11-26

**Authors:** Vytautas Stankus, Petras Navickas, Anžela Slušnienė, Ieva Laucevičienė, Albinas Stankus, Aleksandras Laucevičius

**Affiliations:** 1Department of Physics, Kaunas University of Technology, 44249 Kaunas, Lithuania; 2State Research Institute Centre for Innovative Medicine, 08410 Vilnius, Lithuania; 3Clinic of Cardiac and Vascular Diseases, Faculty of Medicine, Vilnius University, 03101 Vilnius, Lithuania; 4Department of Rehabilitation, Physical and Sports Medicine, Faculty of Medicine, Vilnius University, 03101 Vilnius, Lithuania

**Keywords:** artifact, noise, elimination, heart rate variability, RRI, R-R interval, algorithm

## Abstract

As heart rate variability (HRV) studies become more and more prevalent in clinical practice, one of the most common and significant causes of errors is associated with distorted RR interval (RRI) data acquisition. The nature of such artifacts can be both mechanical as well as software based. Various currently used noise elimination in RRI sequences methods use filtering algorithms that eliminate artifacts without taking into account the fact that the whole RRI sequence time cannot be shortened or lengthened. Keeping that in mind, we aimed to develop an artifacts elimination algorithm suited to long-term (hours or days) sequences that does not affect the overall structure of the RRI sequence and does not alter the duration of data registration. An original adaptive smart time series step-by-step analysis and statistical verification methods were used. The adaptive algorithm was designed to maximize the reconstruction of the heart-rate structure and is suitable for use, especially in polygraphy. The authors submit the scheme and program for use.

## 1. Introduction

Heart rate variability (HRV) studies are based on the instantaneous heart rate time series analysis that uses beat-to-beat RR interval (RRI) data derived from an electrocardiographic (ECG) signal. Over the years, research on heart rate and its variability has gained a lot of traction and has spread not only to medicine but to fields of sports, ergonomics, and personal healthcare as well [1]. As it became widely available to every person who cares about his body’s state and health, various time series analysis methods have been proposed. However, in all cases, the accuracy of the results is highly dependent on artifacts occurring in the time sequence [2,3]. These ECG artifacts inevitably originate due to both internal physiological processes that influence the normal sinus heart rhythm (e. g. respiration) [4], as well as external determinants caused by body movements or power line interference, external electromagnetic fields, and hardware/software failures [5], all of which are called “noise” or “artifacts”. These can be classified as base-line wander [6,7,8], power-line interference [8,9,10], muscle artifacts [7,8,11], and channel noise [12]. Although hardware and software development has made great strides in proposing a wide range of solutions and tools for reducing the number of ECG artifacts, the problem of RRI sequence quality remains relevant to this day [13,14,15], since artifacts representing less than 0.1% of the overall record duration may cause variations of up to 50% in some HRV metrics [16].

To improve the quality of the RRI data, researchers have two options to eliminate artifacts. First, to improve the overall signal quality of an ECG signal and improve the accuracy of the ECG QRS complex recognition during the investigation [17] or the second—to identify artifacts in real time or retrospectively within the recorded RRI sequences [18]. Different methods are used for each of the aforementioned options, as the ECG and RRI signals are different in nature: the ECG signal is continuous and its amplitude is measured in volts, while the RRI signal is discrete in nature and its amplitude is measured in seconds. In addition, when the ECG is converted to digital form, the ECG time axis step is constant, while the RRI time axis step is variable. Consequently, this results in different methods being used to process these signals. Overall ECG artifact identification and removal are made possible by various algorithms and are closely dependent on the aim [19]: a data-driven mechanism of empirical mode decomposition [20,21], deep-learning-based models [22], wavelet-based models [23], and sparsity-based, Bayesian-filter-based, and Hybrid models [24]. The majority of them focus on short-duration artifacts, which are often treated in the same way as ectopic beats [25,26,27]. The proposed gap-filling methods are based on an integrated pulse frequency modulation model [25], a rejection filter on the basis that nonpathological artifacts are of small duration and large amplitude [28], based on the tachogram using cubic splines [29].

However, the previous methods change the overall duration of the recording, are only suitable for isolated artifacts, and have not been evaluated for longer artifact segments. There are algorithms designed for longer gap filling, some of which are using Gaussian-distribution-based methods [30]; unfortunately, they are less common and are not designed to retain the overall recording duration. 

Another very common method of correction is based on the RRI signal value threshold, when RRI values appear beyond the impossible limits; for example, 350 ms and 1715 ms, respectively [31]. After removing the identified erroneous values, they are replaced by interpolated data. The resulting time sequence is usually depicted as amplitude on the abscissa axis, while the ordinary axis represents frequency or real time. This depiction of the discrete signal enforces the investigators to apply common amplitude artifacts removing algorithms to RRI sequences, despite the inadequacy of its nature, since various filters dominate between them. Therefore, applying it enables us to adequately eliminate a small number of artifacts and thus reduce the overall sequence dispersion and its distribution in the frequency axis. However, it ought to be emphasized that even though the entire duration of the RRI sequence in such a situation becomes slightly shortened or lengthened, the removal of artifacts has a low impact on the time domain and frequency domain HRV metrics [31]. 

To summarize, there are algorithms for ectopic beats correction [25,32,33,34,35], restoration of missing heartbeats [36], and noise elimination [37]; however, an entirely different picture is seen in which heart rate registrations occur alongside the other processes (physiological and non-physiological). In this instance, it is necessary to maintain a precise synchronization of each RRI with other measurable processes. Therefore, the use of a traditional filter becomes inadequate for the expected results of the analysis, especially when looking for associations between the heart rate and other internal (electromyography, electroencephalography, and others) as well as external factors [18]. It is well acknowledged that “accurate R–R interval artifact correction and editing methods are needed” [38], especially when RRI sequences are very long, and it is complicated to apply manual methods. Consequently, after encountering this issue when dealing with multidimensional processes, we aimed to propose an original RRI sequence artifact elimination algorithm suited for long-term (hours or days) sequences that would overcome the aforementioned limitation and would preserve the overall RRI sequence temporal structure in real-time recordings.

## 2. Materials and Methods

### 2.1. Materials

The proposed algorithm was evaluated using long RRI time-sequences collected during monitoring with a 3-channel ECG Holter monitor that has a sampling rate of 600 Hz and 12-bit sampling characteristics (Card(X)plore BPM, ECG & Actigraph from Meditech-Ltd., Budapest, Hungary). The algorithm was based on ECG data from 97 individuals. The analysis encompassed 20,000 RRIs of each patient (~5 h). Dedicated software (LabVIEW, National Instrument, Austin, TX, USA) was used to perform the ECG signal analysis and to employ the proposed algorithm. The study was approved by the Vilnius Regional Biomedical Research Ethics Committee and written informed consent was obtained from all participants (No. 158200-17-889-400). Data analysis was performed using the IBM SPSS Software 21.0 version (SPSS, Chicago, IL, USA). The proposed adaptive algorithm was installed on a machine with LabVIEW and Visual C++ 6.0 running Microsoft Windows 10 on a 3.5 GHz processor and 8 GB of memory.

### 2.2. Study Population

In total, 97 middle-aged 50–55-year-old subjects (both men and women) were included in the study. Sixty-three subjects were diagnosed with metabolic syndrome (MetS) (according to the modified National Cholesterol Education Program, Adult Treatment Panel III (NCEP-ATP III) criteria), in all of whom arterial blood pressure (BP) of more than 130/85 mmHg was present or they were treated with antihypertensive drugs. The remaining 34 subjects were the same age without MetS and had BP was less than 130/85 mmHg. All subjects were recruited from the ongoing Lithuanian High Cardiovascular Risk (LitHiR) primary prevention program as a part of an ongoing project that analyses metabolic syndrome patients [39]. In addition, daily activities according to the data from the subjects’ diary are presented in Appendix A.

### 2.3. Proposed Novel Algorithm

We propose a novel adaptive algorithm for noise reduction of the RRI time series, which is aimed at the main limitation that is common for traditionally used algorithms: the alteration of the overall RRI sequence length. The proposed algorithm is designed to maximally restore the sinusoidal heart rate structure and is based on several principles:The difference between the actual duration of the RRI recording and the sum of the identified RRI values at any point in time may not exceed the difference of one average RRI value of the measured interval.The spectral characteristic of the RRI sequence obtained after artifact elimination cannot be artificially distorted.The proposed algorithm is designed for long-term (hours or days) sequences, where it is difficult to precisely carry out artifact elimination without changing the timeline structure.

### 2.4. Procedural Steps

A detailed representation of the algorithm’s processes is given in Figure 1; however, the algorithm can be conditionally divided into the following steps:

#### 2.4.1. Step 0

The first objective is to separate artifacts from normal RRI values. To do this, the ratios of adjacent RRI values (Coef, Figure 1) were compared to the values of selected window maximum and minimum limits. If there is one artifact in the time sequence, the division of adjacent values (before and after artifact) creates two incorrect values. Therefore, each RRI value must be divided by the average of the moving window of three RRI values. The selected window indicates that all new divided values that appeared in the middle of the window remain unchanged, but values, which appeared outside of them, are named as artifacts. Statistical methods should be used to find these criteria. 

#### 2.4.2. Step 1

A very important step in initiating further actions (Init subroutine—Figure 1). A constant memory field (length = 30) filled with RRI values with normal frequency characteristics is formed and averaged. Afterwards, a temporary RRI memory field is allocated for normal and artifact-cleared RRI values with a short moving period (~3 s), from which their moving average will be continuously calculated. It is of the greatest importance that the first three initial RRI values are taken from the sequence itself and without artefacts.

#### 2.4.3. Step 2

All RRI values whose ratio to the moving window average exceeds the upper criterion are corrected with RRI values close to the RRI at that moment (Fill_RR, look at subroutine—Figure 1). The artificial RRI field is filled with RRI values from the backup memory space (length = 30). To avoid the distortion of the frequency sequence structure, the field addresses are chosen randomly. Their average value (length = 30) must be equal to the moving window average (length = 3) at that moment. The time difference between the sum of the bad RRIs found and the sum of the time values filled with new values is stored in the buffer memory. The resulting number of RRIs in the sequence increases. To preserve the time equivalence of the last steps of the observed structure, this step can be continued indefinitely. The added RRI frequency characteristic is equivalent to white noise; it will not affect the other results and will not change their average. The RRI signal values are uncorrelated (independent), just like white noise. The mean of the white noise values is zero and the frequency response is flat and covers the entire frequency band. Thus, by filling the noise-affected sections with a sequence of values whose properties correspond to those of white noise, the characteristics of the RRI signal are preserved. If an artefact exceeds the value of only one RRI, it can be replaced by an average of the moving window values or a predictive value derived from a selected short section of the RRI sequence. 

#### 2.4.4. Step 3

RRI values, which have a ratio to the moving window average below the minimum criterion, are corrected by summing the value of the artefact first with the value of the buffer and then with the subsequent values in the time sequence up to the value of the RRI that exceeds the level of the moving field average. It may even be necessary to aggregate several short time intervals, which reduces the number of RRIs in the sequence. Consequently, the shorter RRI value(s) are replaced by the “correct” RRIs values, which are based on the moving window average (or prediction), and the difference of the resulting time sum with it is placed back in the buffer memory.

#### 2.4.5. Step 4

The RRI values, which ratios with the average of a window are within the selected coefficients, are considered normal and are not changed. In this step, the buffer value is continuously checked. When this value exceeds the average of the moving window after the next steps, a new RRI value is added to the main stored heart rate field and the time sequence number is extended by one additional value. The buffer value decreases.

#### 2.4.6. Step 5

After steps 2, 3, and 4, the normal or corrected RRI value(s) are stored in memory or in a file and summed. In this way, this and the following steps maintain the equality between the sum of the original (or real-time) RRI and the sum of all the corrected RRIs.

### 2.5. Programing Step

With the addition of the necessary inserts, the simplified program provided in Appendix B could be used. The authors deliberately retained a similar symbolism of parameters in the program block diagram and program functions as far as possible.

## 3. Results

### 3.1. Criteria Selection

In most cases, RRI values that are outside the distribution of three standard deviations (SD) from the mean (more or less ±3 × SD) are attributed to artifacts. This criterion can be applied when the sequence distribution is normal (Gaussian). However, the distribution of the obtained values changes after the division of adjacent RRI values. It is known that the ratio of two normally distributed random variables has a Cauchy distribution [40]. Hence this distribution has no instantaneous values: mean, variance, or higher moments defined. The mean of the sequence after the division is close to one. In order to find the required criteria, pre-whitening is often performed in such cases, which can be achieved using the autoregressive (AR) model [41]. Since the main one-step operation of artifact exclusion is the division of adjacent RRI values, we consider that in this case, the application of the first-order AR model to RRI sequences is guaranteed by the removal of the first-order link between adjacent RRI. 

We use the following dependencies to solve the problem:(1)$$e_{n}=RRI_{n}+a_{1}RRI_{n-1}$$
where $e_{n}$ is the Gaussian white noise series with a mean of non-zero and $a_{1}$ is the first-order autoregressive coefficient. Remaining noise standard deviation:(2)$$\sigma_{e}=\sqrt{\frac{1}{N-1}\sum_{n}{\left(e_{n}-\bar{e}\right)}^{2}}$$
where $\bar{e}$ is the mean of *e_n_*, *e*(*n*) is the Gaussian white noise series with a mean of non-zero; *a*(1) is the first-order autoregressive coefficient; *E*[*x*] is the expected *x* value of the mean.

The AR model was applied to the sequence that results from dividing each RRI value by the moving average of the previous three values. The noise sequence obtained after autoregressive filtering is close to a normal distribution, from which it is possible to find its mean and to find confidence limits of 0.997. These coefficients were used to distinguish normal RRIs from elongated “top” and truncated “bottom” artificial values, which were eliminated. The value three times the standard deviation in both directions of this resulting noise sequence becomes the window limits for the time sequence. The mean of adjacent RRIs ratios is equal to one; therefore, the criteria will be equal to 1 ± 3 × SD. The maximum criterion is obtained by adding 3 × SD value to mean and minimum—by subtracting 3 × SD from it. Outside of these 99.7% limits, RRI values are classified as artifacts.

### 3.2. Criteria Check

The following methods were used to estimate the effectiveness of the presented algorithm. The standard time-domain variability measurements of values before (RRIb) and after (RRIa) noise elimination are as follows:(3)$$\mu =\frac{1}{N-1}\sum_{n}RRI_{n}$$
where *µ* is the mean of *RRI*, *N—*number of *RRI* values. 

Variance of *RRI*:(4)$$\sigma^{2}=\frac{\sum_{n}RRI_{n}^{2}}{N-1}-{\left[\frac{\sum_{n}RRI_{n}}{N-1}\right]}^{2}$$

Difference of *RRI* variance (before–after):(5)$$\Delta \sigma^{2}=\sigma_{RRIb}^{2}-\sigma_{RRIa}^{2}$$

Adjacent differences of *RRI* series:(6)$$\Delta RRI_{n}=RRI_{n}-RRI_{n-1}$$
for *RRIa* and *RRIb*.

Variance of the adjacent differences for Δ*RRIa* and Δ*RRIb*:(7)$$\sigma_{\Delta RRI}^{2}=\frac{\sum_{n}\Delta RRI_{n}^{2}}{N-1}-{\left[\frac{\sum_{n}\Delta RRI_{n}}{N-1}\right]}^{2}$$

Short-term standard deviation:(8)$$\sigma 1=\sqrt{\frac{\sigma_{\Delta RRI}^{2}}{2}}$$

Long-term standard deviation:(9)$$\sigma 2\;=\sqrt{2\cdot \sigma^{2}-\Delta \sigma_{\Delta \mathrm{RRI}}^{2}}$$

RRI total time difference:(10)$$tRRI=\sum_{n}RRIb_{n}-\sum_{n}RRIa_{n}$$

Numbers of normal RRIs and artifacts with extended (incRRI) and shortened (decRRI) intervals identified.

#### 3.2.1. Time Frequency Analysis

After centering both sequences (subtracting the mean from the sequence), the AR coefficients were calculated by minimizing the least square errors of the forward and backward predictions. AR coefficients return to the estimated coefficients with the value of AR order (*k* ≥ 16) [42]. The following parameters were calculated in both time sequences (before and after the artifact elimination):

Residual noise before *e*1*_n_* and after artifact elimination *e*2*_n_* according to functions:(11)$$e_{n}=RRI_{n}+a_{1}RRI_{n-1}+\ldots +a_{k}RRI\left(n-k\right)$$

Remaining noise dispersion:(12)$$\sigma_{e}^{2}=\frac{\sum_{n}e_{n}^{2}}{N}-{\left[\frac{\sum_{n}e_{n}}{N}\right]}^{2}$$
where *e_n_* is the Gaussian white noise series with a mean of zero. 

Degree of noise reduction in percent:(13)$$\Delta \sigma_{e}^{2}=100\cdot \frac{\sigma_{e1}^{2}-\sigma_{e2}^{2}}{\sigma_{e1}^{2}}$$

Among the subjects, some experienced both fast and slow heart rates (their RRI averages ranged from 656 ms to 1255 ms). The mean of overall RRI was equal to 891 ± 118 ms and did not change after artifact elimination (Table 1). The data presented in the table shows that the variability indexes have significantly changed after the artifact elimination. The total size of the dispersion after the calculations has also significantly decreased (*p* < 0.001). 15–16,066 ms^2^ RRI variance interval difference was obtained. This change was also confirmed by other heart rate variability indexes: the variance of adjacent RRI (*p* < 0.001) and values of the Poincare plot (*σ*1 and *σ*2) (*p* < 0.001). The residual white noise level decreased by 65.0 ± 21.0%, from 2643 ± 3374 ms^2^ to 473 ± 288 ms^2^ (*p* < 0.001). We did not compare the power spectral density of AR spectral density before and after the artifact elimination because, in all cases, as shown in the table, the variance of the two sequences varied greatly. It is clear that the coefficients of AR also differ for the same reasons. In addition, the algorithm was validated in RRI sequences where the total number of artifacts was known. They were not large and were repaired. Accordingly, after artifact elimination, the total number of RRIs increased by up to 230 (mean = 38 ± 51) or decreased by up to 576 RRIs (mean = 45 ± 84) (Table 1). Compared to the entire length of the sequence (20,000) this is a small number, but it has greatly changed the sequence variance.

#### 3.2.2. Time Complexity Analysis

The used algorithm is based on the length of the adjacent RRI. Normal RRI values should be higher than the minimum criteria and lower than the maximum. On this basis, the normal value of RRI is separated from the artifact. For the determination of these criteria, the ratios between all adjacent RRI values were calculated. A statistical analysis of the distribution of these ratios was carried out and their histograms were monitored. Particular attention has been paid to the magnitude of the mutual limits used in the artifact-finding algorithm.

Ninety-seven case studies showed that this value (Table 1, 3 × SD of the ratio of adjacent RRI) varied around 1 from 0.05 to 0.64 with an average of 0.18 ± 0.11. Figure 2 shows that the 3 × SD value is distributed around this average. This indicates that the upper value of the criterion is very close to 1.18 ± 0.11, and the lower value is 0.82 ± 0.11 (1.0 − 0.18 = 0.82). Before the beginning, the artifact elimination procedure of the first-order AR simulation must be performed, and the selection of the exact coefficients is required (Figure 2). The distribution of moments of noise generated by AR simulation is asymmetric and depends on the number of artifacts (Figure 2 and Table 1). Visually, in a separate example, the noise distribution is clearly visible (Figure 3). The values observed on both sides of the average are attributed to artifacts (±3 × SD = 0.18 ± 0.11).

## 4. Discussion

There are a plethora of fundamentally different RRI processing methods being used, which sometimes makes comparing different HRV data difficult. Although many HRV software uses automatic repair systems, they usually lack information about how they behave with artifacts. Two basic principles were used to construct our proposed method. The first is that:1.Autoregressive research and other methods show that strong relationships in the RRI sequence remain not only between adjacent RRIs but also extend to the number of RRI values from 4 to 50 (order of AR model) [43,44].

In that period, any noise will be non-stationary and very strong. Comparing artifacts with adjacent RRI values allows us to identify them within the time sequence. Therefore, changes in adjacent RRI values in our work have been used as the main indicators for separating artifacts from physiological RRI changes. A minimum of three moving RRI values (for average and for window) were selected, with which the next RRI was compared. The stability of the algorithm is conditioned by the fact that none of the artifacts found in all sequences can pass to the moving average window. The last value (third) must include only normal or corrected RRI values in the scrolling field. The second principle is based on the perception that:
2.Each RRI sequence is made up of “fragmented” real-time intervals, the sum of which should correspond to the actual elapsed registration time.

The proposed algorithm follows the principle that the duration of the time sequence should remain identical to the actual elapsed time, independent of normal RRI values and the number of artefacts present: therefore, long-duration artefacts should be filled with similar expected duration values and very short duration artefacts should be aggregated to a value close to the former RRI (using the above-described method). Changing the length of long or short time intervals by moving the selected average window may shorten the overall sequence duration and lead to a duration difference compared to the actual elapsed time. Therefore, the time difference between the original RRI sequence and the one after artifact elimination, which is always shorter, is placed in the backup buffer and used in the next stage of the operation. Compared to other popular methods, our proposed algorithm does not change the overall length of the ECG recording. Following the use of this algorithm in the analysis, one can safely apply methods of multivariate spectral analysis, which require complete synchronization between the processes registered, since only under such conditions it is possible to accurately assess the existing relationships between the observed processes.

Nonetheless, there are some limitations that are important to note. Firstly, due to the fundamental difference of the proposed method, we were not able to directly compare it with other methods, which are being used by the majority of authors—not even if we used a simple method such as mean-square error: currently, the most commonly used classical artifact elimination algorithm, which maintains a constant number of RRI in a sequence, eliminated artifacts by increasing RRI sequence length in one place and shortening it in another. Secondly, we acknowledge that by design, the proposed algorithm is suited for long-term (hours or days) sequences and, therefore, is not applicable in very short sequences.

## 5. Conclusions

A novel artifact elimination algorithm that does not affect the overall structure of the RRI sequence and does not alter the overall duration of the sequence has been proposed, and its effectiveness with real-life data has been presented. The proposed algorithm is designed for long-term (hours or days) sequences, in which it is difficult to precisely carry out artifact elimination without changing the timeline structure, and it is especially suitable for use in polygraphy when multiple physiological and other processes associated with other cardiac arrhythmias are registered in parallel. Following the use of this algorithm in the analysis, one can safely apply methods of multivariate spectral analysis, which require a complete synchronization between the processes registered, since only under such conditions is it possible to accurately assess the existing relationships between the observed processes.

## Figures and Tables

**Figure 1 sensors-22-09213-f001:**
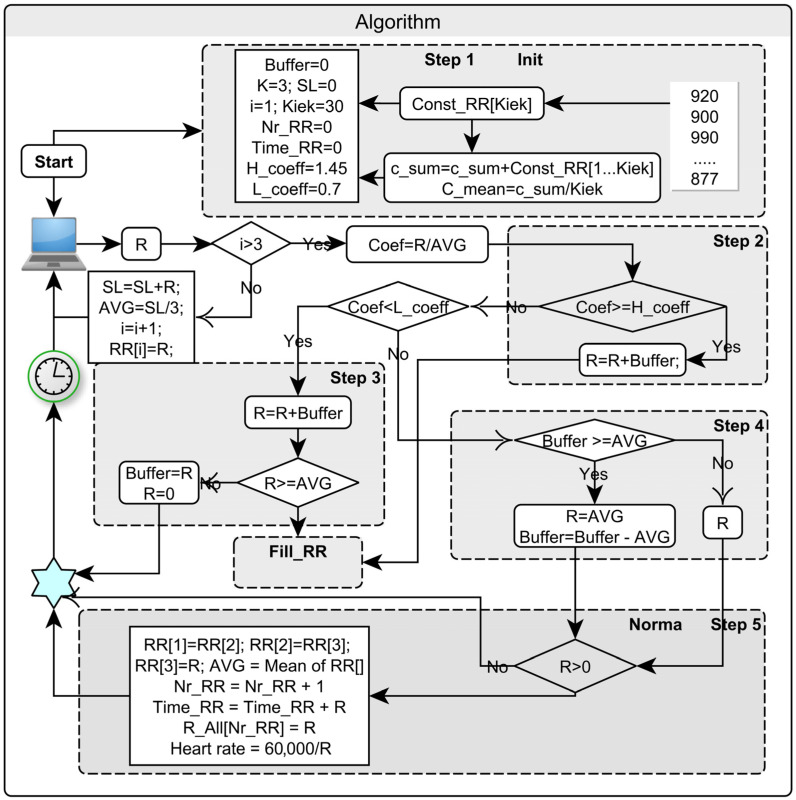
The stepwise procedure of adaptive algorithm (R—input of RR interval from file or device; AVG—average of 3 RR intervals; Coef—division of RRI and average; L_coeff and H_coeff—minimal and maximal criteria; RR[ ]—array of normal RRI; Time_RR—counter of time; Buffer—reserve of RR interval).

**Figure 2 sensors-22-09213-f002:**
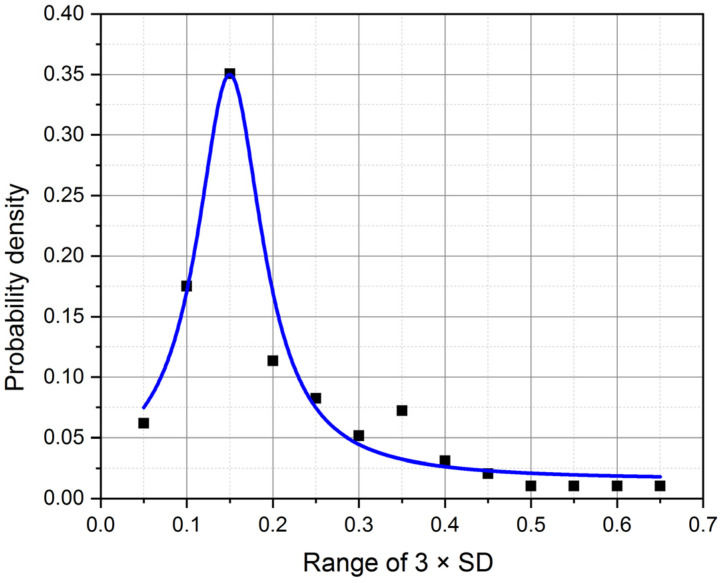
Distribution of averaged artifact standard deviation (3 × SD) of all 97 cases.

**Figure 3 sensors-22-09213-f003:**
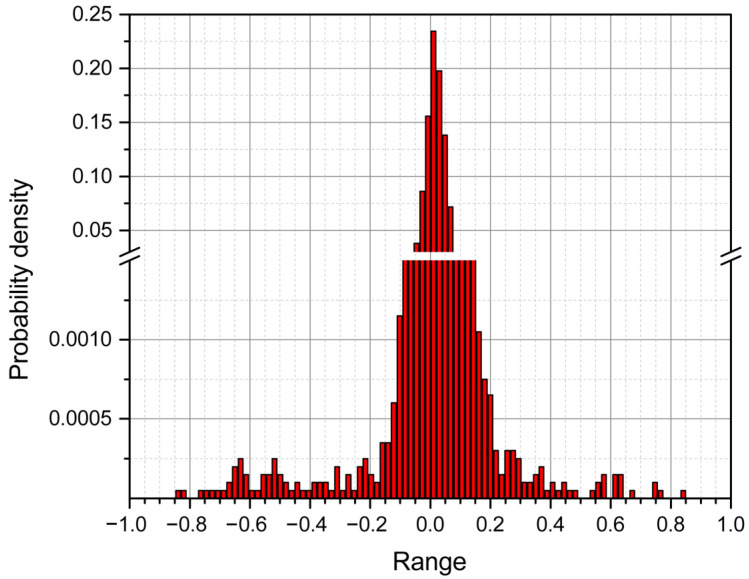
Artifact distribution after autoregressive RRI modeling in a separate case.

**Table 1 sensors-22-09213-t001:** Changes in RRI variability before and after artifact elimination.

Variable, Units	Variable Symbols	Before		After		Paired *t*-Test
		Range	Mean ± SD	Range	Mean ± SD	
Mean of RRI, ms	*μ*	656–1255	891 ± 118	656–1258	890 ± 119	1.79
Variance of RRI, ms^2^	*σ* * ^2^ *	2736–45,450	14,247 ± 8586	2721–44,277	12,298 ± 7556	6.57 *
Difference of RRI variance, ms^2^	Δ*σ* *^2^*			15–16,066	1948 ± 2922	
Variance of adjacent RRI, ms^2^	$\sigma_{\Delta RRI}^{2}$	83–29,395	3959 ± 5481	68–2561	656 ± 481	6.07 *
Variance of AR noise, ms^2^	$\sigma_{e}^{2}$	77–17,174	2643 ± 3374	60–1469	473 ± 288	6.53 *
Decrease in variance AR noise, %	$\Delta \sigma_{e}^{2}$			13–98	65 ± 21	
RRI total time difference, ms	*tRRI*			−1596–968	215 ± 430	
Short-term, ms	*σ*1	6–121	37 ± 24	6–36	17 ± 6	9.29 *
Long-term, ms	*σ*2	74–299	156 ± 46	74–297	149 ± 44	6.78 *
Normal RRI number	Nr			19,263–19,999	19,915 ± 118	
Increased RRI number	incNr			0–230	38 ± 51	
Decreased number of RRIs	decNr			0–576	45 ± 84	
3 × SD of ratio of adjacent RRI		0.05–0.64	0.18 ± 0.11	0.02–0.04	0.051 ± 0.025	11.8 *

* denotes a statistically significant difference (*p* < 0.001). Abbreviations: RRI—R-R interval; AR—autoregressive.

## Data Availability

Not applicable.

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
