# Peer review of "A Novel Adaptive Noise Elimination Algorithm in Long RR Interval Sequences for Heart Rate Variability Analysis"

_sensors, 2022, doi:10.3390/s22239213_

Round 1

Reviewer 1 Report (Previous Reviewer 2)

Unfortunately, the authors still have not considered all my comments. Mainly, the authors extended the introduction and the references. Unfortunately, they did not consider the critical substantive comments that are necessary to be able to assess the usefulness of the proposed solution. Therefore, I renew my comments:

 1. Why was the focus only on white noise? What is the reason why this type of hype is dominant in practice? What about violet, pink, blue, brown noise (etc)? How will the proposed approach be effective for this type of noise?

2. What were the test conditions during the acquisition of the ECG signal? What activities did the subjects undertake (it is necessary to assess the effectiveness of the reduction of movement artifacts)? Was the signal chain exposed to electro-magnetic field? What kind of noise can have occurred during the test? What is the characteristics of the research sample (age, health, etc.)?

3. The verification criteria are not clearly defined in the article.

4. In order to assess the usefulness of the proposed approach, it is worth verifying the proposed approach by comparing it (compare research results for authors proposition and proposition other people) with other methods that reduce noise and artifacts. There are many methods of this type, for example: [a] - [f]. It is worth noting that for the comparison it is worth using the noise reduction methods and artifacts that are used for the ECG signal and for the PPG signal, because in both of these signals (non-stationary signals sensitive to noise, interference and artifacts) the principle of noise reduction techniques is analogous.

5. I also propose to add nomenclature (description of used symbols and abbreviations) to improve the readability of the article.

6. Equations should be written using mathematical symbols. Each function should be defined beforehand (e.g., SUM). 

[a] Kher R., Signal Processing Techniques for Removing Noise from ECG Signals, Journal of Biomedical Engineering and Research, vol. 3, art. no. 101, pp. 1-9, 2019.

[b] Chatterjee, S., Thakur, R.S., Yadav, R.N., Gupta, L. and Raghuvanshi, D.K., Review of noise removal techniques in ECG signals, IET Signal Process., vol. 14, pp. 569-590, 2020.

[c] Dwivedi, A.K., Ranjan, H., Menon, A. et al. Noise Reduction in ECG Signal Using Combined Ensemble Empirical Mode Decomposition Method with Stationary Wavelet Transform, Circuits Syst Signal Process, vol. 40, pp. 827–844, 2021.

[d] Jarchi D, Charlton P, Pimentel M, Casson A, Tarassenko L, Clifton DA, “Estimation of respiratory rate from motion contaminated photoplethysmography signals incorporating accelerometry”, Healthcare Technology Letter, vol. 6, no. 1, pp. 19-26, 2019.

[e] Kuwalek P., Burlaga B., Jesko W., Konieczka P.,”Research on methods for detecting respiratory rate from photoplethysmographic signal”, Biomedical Signal Processing and Control, vol. 66, art. no. 102483, 2021.

[f] Zheng X., Dwyer V.M., Barrett L.A., Derakhshani M., Hu S.,”Adaptive notch-filtration to effectively recover photoplethysmographic signals during physical activity”, Biomedical Signal Processing and Control, vol. 72, Part A, art. no. 103303, 2022.

Author Response

Reviewer 1

Unfortunately, the authors still have not considered all my comments. Mainly, the authors extended the introduction and the references. Unfortunately, they did not consider the critical substantive comments that are necessary to be able to assess the usefulness of the proposed solution. Therefore, I renew my comments:

Dear reviewer, firstly we would like to thank you for your valuable time reviewing the article as well as your useful insights.

  1. Why was the focus only on white noise?

Lines 156,157 answer this question: “To avoid the distortion of the frequency sequence structure, the field addresses are chosen randomly.” In addition, lines 162, 163: “The added RRI frequency characteristic is equivalent to white noise, it will not affect the other results and will not change their average.” In order to justify our choice we supplemented the following explanation: “The RRI signal values are uncorrelated (independent), just like white noise. The mean of the white noise values is zero and the frequency response is flat and covers the entire frequency band. Thus, by filling the noise-affected sections with a sequence of values whose properties correspond to those of white noise, the characteristics of the RRI signal are preserved.”

What is the reason why this type of hype is dominant in practice?

The White noise PowerDensityFunction is in many cases a convenient mathematical idealization of a real stochastic process. The RRI signal is also a stochastic process. In the case of long-term signals, it is the characteristics of the whole signal that are important, not a single value taken in isolation, and it is therefore important to fill the noise-affected sections with a sequence of expressions that do not change the properties of the signal (in this case the RRI signal) under analysis. The individual RRI values used for filling must fall within the range of 350 ms and 1715 ms defined by human physiology [31].

What about violet, pink, blue, brown noise (etc)?

Colour noise samples correlate and their frequency characteristics do not coinside with RRI data frequency characteristics. Thus, correcting noise-affected RRI samples with values corresponding to the colour noise characteristics distorts/modifies the RRI signal characteristics. This is simply not acceptable.

How will the proposed approach be effective for this type of noise?

Neither the type of noise nor the nature of the noise is relevant to the effectiveness of the proposed algorythm.

  1. What were the test conditions during the acquisition of the ECG signal? What activities did the subjects undertake (it is necessary to assess the effectiveness of the reduction of movement artifacts)? Was the signal chain exposed to electro-magnetic field? What kind of noise can have occurred during the test? What is the characteristics of the research sample (age, health, etc.)?

One of the objectives of the algorithm's application was not to restrict its use depending on the nature of daily activities: working, walking, driving, eating and lying in bed. The types of measured activities and their duration is presented in the supplementary Table 1. Data were collected on the basis of a detailed diary provided by the subjects.

Table 1. Daily activities according to the data from the subjects' diary

Daily activities, in hours

Subjects without MetS

Subjects with MetS

Mean ± SD

Mean ± SD

Working

4.48 ± 4.08

3.94 ± 3.64

At home

2.12 ± 1.78

2.30 ± 2.18

Walking

1.22 ± 0.98

1.54 ± 1.76

Driving

0.71 ± 0.84

1.08 ± 1.18

Eating

1.49 ± 0.88

0.98 ± 0.45

Lying in bed

2.35 ± 2.23

1.90 ± 2.06

Abbreviations: MetS – metabolic syndrome; SD – standard deviation

One of the study conditions for the subjects was to avoid strong electromagnetic radiation stations, electrical devices, not to touch large size conductors (elevators cabins metal plates, metal parapets surfaces and etc.). In addition, subjects were required to wear cotton T-shirts during the research and to keep a distance from other technical equipment.

  1. The verification criteria are not clearly defined in the article.

We have tried to clarify the verification criteria, which in part are defined in section “3.1 Criteria selection” as well as in Table 1, which provides all the information on how the variables have evolved after the artifacts elimination. We believe that an additional separate section is not necessary given that we do not present a comparison with other techniques.

  1. In order to assess the usefulness of the proposed approach, it is worth verifying the proposed approach by comparing it (compare research results for authors proposition and proposition other people) with other methods that reduce noise and artifacts. There are many methods of this type, for example: [a] - [f]. It is worth noting that for the comparison it is worth using the noise reduction methods and artifacts that are used for the ECG signal and for the PPG signal, because in both of these signals (non-stationary signals sensitive to noise, interference and artifacts) the principle of noise reduction techniques is analogous.

In response to your comment, it should be noted that the method proposed in the paper is not designed to eliminate noise from the ECG (bioelectrical heart signal) signal, but to eliminate artifacts from the RRI sequence ("used noise elimination in RRI sequences"). Since these are fundamentally different signals are different in nature. ECG signal amplitude values are measured in volts and RRI signal amplitude values are measured in milliseconds (ECG signal is continuous in nature and RRI signal is discrete in nature). In addition, the time axis step of the RRI discrete signal is not constant. The sum of the ECG amplitude values has no physical meaning, whereas the sum of the amplitudes of the RRI signal is equal to the recording duration of the ECG signal. These signals (RRI, ECG) have different intrinsic properties and therefore require different processing methods than the ones you mentioned [a, b, c, d, e f].

We emphasise that the unique feature of the proposed method is that it preserves not only the temporal-frequency properties of RRI, but also the original RRI registration duration and average (“it will not affect the other results and will not change their average”). It is also worth highlighting that artifacts representing less than 0.1% of the overall record duration may cause variations of up to 50% in some heart rate variability (HRV) metrics [16], which in part makes comparing

There's a whole range of work aimed to remove noise from the RRI signal:

[1] Benchekroun M, Chevallier B, Istrate D, Zalc V, Lenne D. Preprocessing Methods for Ambulatory HRV Analysis Based on HRV Distribution, Variability and Characteristics (DVC). Sensors (Basel). 2022 Mar 3;22(5):1984. doi: 10.3390/s22051984. PMID: 35271128; PMCID: PMC8914897.

[2] Barnaby D, Ferrick K, Kaplan DT, Shah S, Bijur P, Gallagher EJ. Heart rate variability in emergency department patients with sepsis. Acad Emerg Med. 2002 Jul;9(7):661-70. doi: 10.1111/j.1553-2712.2002.tb02143.x. PMID: 12093705.

[3] Griffin MP, Moorman JR  Toward the early diagnosis of neonatal sepsis and sepsis-like illness using novel heart rate analysis. Pediatrics, 2001 107:97–104

[4]  Salo MA, Huikuri HV, Seppänen T. Ectopic beats in heart rate variability analysis: effects of editing on time and frequency domain measures. Ann Noninvasive Electrocardiol. 2001 Jan;6(1):5-17. doi: 10.1111/j.1542-474x.2001.tb00080.x. PMID: 11174857; PMCID: PMC7027677.

These works typically address two challenges:

1) the task of identifying the start and end of a noise-affected RRI signal segment (identification RRI outliers intervals);

2) the task of correcting this segment (how to manage outliers once they have been identified) in such a way that no change in the RRI characteristics occurs.

However, unlike our work, these papers do not address the fact that the duration of the "corrected" RRI signal (when the x-axis is time /*P.S. if the x-axis is time, then the time axis step is not constant*/) or the sum of the amplitude values (when the number of the RRI is presented on the x-axis) does not coincide with the duration of the ECG recording (measured in seconds). This aspect is important for RRI analysis [Catai AM, Pastre CM, Godoy MF, Silva ED, Takahashi ACM, Vanderlei LCM. Heart rate variability: are you using it properly? Standardisation checklist of procedures. Braz J Phys Ther. 2020 Mar-Apr;24(2):91-102. doi: 10.1016/j.bjpt.2019.02.006. Epub 2019 Feb 26. PMID: 30852243; PMCID: PMC7082649]. On the other hand, this aspect is particularly important when multi-channel recording of biomedical signals (polygraphy) is used to address the questions raised by the study when several signals (ECG, PPG, EEG, BCG, etc..) are recorded simultaneously.

 /* Explanation.

As no other authors have compared the durations of ECG and "corrected" RRI signals, it is not possible to answer the question, which of the known methods is better in terms of preserving the original signal duration.

  1. I also propose to add nomenclature (description of used symbols and abbreviations) to improve the readability of the article.

Thank you for the suggestion. A nomenclature and abbreviations list has been provided as appendix C.

  1. Equations should be written using mathematical symbols. Each function should be defined beforehand (e.g., SUM).

We are grateful for the suggestion. The mathematical equations have been rewritten.

Thank you once again for your comprehensive review. We tried to address your comments as much as possible and correct the article accordingly. We hope that you will find these corrections sufficient.

Reviewer 2 Report (New Reviewer)

Summary:

The authors present a heuristic method to eliminate artifacts in long duration R-to-R interval sequences and validate the method in 97 middle aged subjects.

General concept comments: 

The article needs to make it clear upfront that the proposed algorithm is suited for long-term sequences and not applicable for short-term HRV”. I suggest the authors revise the title and abstract reflect this. One of the major shortcomings of the work is lack of comparative studies. While the authors attribute this to the fundamental difference of the proposed method (the method does not change the overall length of the ECG recording), it is not fully clear why a comparison similar to Table 1 cannot be performed against state-of-the-art methods. The authors should provide more details about the ECG Holter monitor and RR intervals collected in the study (what is the ECG sampling frequency, how were the QRS complexes detected, what is the QRS peak detection accuracy, etc). It is not clear why the authors chose subject groups with or without metabolic syndrome for validation. If this is relevant, then sub-group analysis should be provided for Table 1. Overall, the work is scientifically sound with appropriate study design and validation, and I recommend accepting the article after the comments are satisfactorily addressed.

Author Response

Reviewer 2

“Summary:

The authors present a heuristic method to eliminate artifacts in long duration R-to-R interval sequences and validate the method in 97 middle aged subjects.

General concept comments:

The article needs to make it clear upfront that the proposed algorithm is suited for long-term sequences and not applicable for short-term HRV”. I suggest the authors revise the title and abstract reflect this. One of the major shortcomings of the work is lack of comparative studies. While the authors attribute this to the fundamental difference of the proposed method (the method does not change the overall length of the ECG recording), it is not fully clear why a comparison similar to Table 1 cannot be performed against state-of-the-art methods. The authors should provide more details about the ECG Holter monitor and RR intervals collected in the study (what is the ECG sampling frequency, how were the QRS complexes detected, what is the QRS peak detection accuracy, etc). It is not clear why the authors chose subject groups with or without metabolic syndrome for validation. If this is relevant, then sub-group analysis should be provided for Table 1. Overall, the work is scientifically sound with appropriate study design and validation, and I recommend accepting the article after the comments are satisfactorily addressed.”

Dear reviewer, firstly we would like to thank you for your valuable time reviewing the article as well as your useful suggestions.

We would like to note that we’ve taken into account your first remark regarding the long-term aspect of the algorithm and we have corrected both the title/abstract as well as the introduction section in order to emphasize the “long-term” aspect.

Furthermore, we have expanded the methods section and provided more details about the ECG acquisition device and ECG analysis methodology.

Finally, we would like to clarify the selection of the study population. Initially, we assessed HRV peculiarities of metabolic syndrome patients (one of the publications: 10.3390/medicina55100700) and came across various artifact elimination issues. As a result, we continued working with the accumulated data. We believe that in terms of algorithm validation such comorbidities ought not to have a significant effect. Nonetheless, we acknowledge that this feature might seem confusing for some readers, therefore, we have rewritten a paragraph in the methods section in order to emphasize the study population selection process and previous works.

Thank you once again for your comprehensive review. We tried to address your comments as much as possible and correct the article accordingly. We hope that you will find these corrections sufficient.

Round 2

Reviewer 1 Report (Previous Reviewer 2)

I am fully satisfied with the corrections and answers to my doubts. I think that the article can be published after some editing corrections. Namely, the equations still contain imprecise functionals (e.g. Var, SD, etc.). In addition, it is worth specifying the methodological differences between ECG and RRI signal studies in the introduction.

Author Response

"I am fully satisfied with the corrections and answers to my doubts. I think that the article can be published after some editing corrections. Namely, the equations still contain imprecise functionals (e.g. Var, SD, etc.). In addition, it is worth specifying the methodological differences between ECG and RRI signal studies in the introduction."

Dear reviewer, thank you once again for your valuable time reviewing the article as well as your insightful suggestions.

“Namely, the equations still contain imprecise functionals (e.g. Var, SD, etc.).”

Thank you for highlighting the remaining imprecisions. We have carefully revised the equations and made the necessary corrections.

In addition, it is worth specifying the methodological differences between ECG and RRI signal studies in the introduction.”

We fully agree with the suggestion to emphasise the methodological differences between ECG and RRI signal analysis. Therefore, we have corrected the introduction section accordingly.

Thank you once again for your comprehensive review. We hope that you will find these corrections acceptable.

This manuscript is a resubmission of an earlier submission. The following is a list of the peer review reports and author responses from that submission.

Round 1

Reviewer 1 Report

Respected Authors,

This paper proposed and demonstrated the efficacy of a novel artifact reduction technique that preserves the RRI sequence's original structure and duration while still removing unwanted artifacts.

Before I get started, I want you to know that I acknowledge the time and energy you spent researching this.

However, there is, in my opinion, it is unclear to me how this new approach is innovative. I believe that your articles for publication in a journal should be reconsidered.

Regards,

Reviewer 2 Report

The research results presented in the paper under review can be interesting for a certain part of the science society. In my opinion, the subject matter of the article is very important and interesting. Hence, it is important to look for methods to reduce noise and artifacts so as to increase the accuracy of feature extraction from biological signals. The method presented in the article can be useful for the indicated purpose. However, I have a few comments. I need an answer to it in order to finally make a decision about the usefulness of the proposed method.

Comments:

1.) I would like to suggest authors to extension of the introduction so that this section can fully present the current state of art.

2.) I also propose to add nomenclature (description of used symbols and abbreviations) to improve the readability of the article.

3.) Equations should be written using mathematical symbols. Each function should be defined beforehand (e.g., SUM).

4.) Although the topic and research results presented in the paper is current and interesting, the references section in the peer-revied paper is poor. The references section does not contain the current state of knowledge in the scope presented in the peer-revied paper.

5.) Why was the focus only on white noise? What is the reason why this type of hype is dominant in practice? What about violet, pink, blue, brown noise (etc)? How will the proposed approach be effective for this type of noise?

6. What were the test conditions during the acquisition of the ECG signal? What activities did the subjects undertake (it is necessary to assess the effectiveness of the reduction of movement artifacts)? Was the signal chain exposed to electro-magnetic field? What kind of noise can have occurred during the test? What is the characteristics of the research sample (age, health, etc.)?

7. The verification criteria are not clearly defined in the article.

8. In order to assess the usefulness of the proposed approach, it is worth verifying the proposed approach by comparing it with other methods that reduce noise and artifacts. There are many methods of this type, for example: [a] - [f]. It is worth noting that for the comparison it is worth using the noise reduction methods and artifacts that are used for the ECG signal and for the PPG signal, because in both of these signals (non-stationary signals sensitive to noise, interference and artifacts) the principle of noise reduction techniques is analogous.

[a] Kher R., Signal Processing Techniques for Removing Noise from ECG Signals, Journal of Biomedical Engineering and Research, vol. 3, art. no. 101, pp. 1-9, 2019.

[b] Chatterjee, S., Thakur, R.S., Yadav, R.N., Gupta, L. and Raghuvanshi, D.K., Review of noise removal techniques in ECG signals, IET Signal Process., vol. 14, pp. 569-590, 2020. 

[c] Dwivedi, A.K., Ranjan, H., Menon, A. et al. Noise Reduction in ECG Signal Using Combined Ensemble Empirical Mode Decomposition Method with Stationary Wavelet Transform, Circuits Syst Signal Process, vol. 40, pp. 827–844, 2021.

[d] Jarchi D, Charlton P, Pimentel M, Casson A, Tarassenko L, Clifton DA, “Estimation of respiratory rate from motion contaminated photoplethysmography signals incorporating accelerometry”, Healthcare Technology Letter, vol. 6, no. 1, pp. 19-26, 2019.

[e] Kuwalek P., Burlaga B., Jesko W., Konieczka P.,”Research on methods for detecting respiratory rate from photoplethysmographic signal”, Biomedical Signal Processing and Control, vol. 66, art. no. 102483, 2021.

[f] Zheng X., Dwyer V.M., Barrett L.A., Derakhshani M., Hu S.,”Adaptive notch-filtration to effectively recover photoplethysmographic signals during physical activity”, Biomedical Signal Processing and Control, vol. 72, Part A, art. no. 103303, 2022.